# LONG-TAIL LEARNING VIA LOGIT ADJUSTMENT

**Aditya Krishna Menon, Sadeep Jayasumana, Ankit Singh Rawat, Himanshu Jain**
**Andreas Veit & Sanjiv Kumar**
Google Research
New York, NY
`{adityakmenon,sadeep,ankitsrawat,himj,aveit,sanjivk}@google.com`

## ABSTRACT

Real-world classification problems typically exhibit an *imbalanced* or *long-tailed* label distribution, wherein many labels have only a few associated samples. This poses a challenge for generalisation on such labels, and also makes naïve learning biased towards dominant labels. In this paper, we present a statistical framework that unifies and generalises several recent proposals to cope with these challenges. Our framework revisits the classic idea of *logit adjustment* based on the label frequencies, which encourages a large *relative margin* between logits of rare positive versus dominant negative labels. This yields two techniques for long-tail learning, where such adjustment is either applied post-hoc to a trained model, or enforced in the loss during training. These techniques are statistically grounded, and practically effective on four real-world datasets with long-tailed label distributions.

## 1 INTRODUCTION

Real-world classification problems typically exhibit a *long-tailed* label distribution, wherein most labels are associated with only a few samples (Van Horn & Perona, 2017; Buda et al., 2017; Liu et al., 2019). Owing to this paucity of samples, generalisation on such labels is challenging; moreover, naïve learning on such data is susceptible to an undesirable bias towards dominant labels. This problem has been widely studied in the literature on learning under *class imbalance* (Kubat et al., 1997; Chawla et al., 2002; He & Garcia, 2009), and the related problem of *cost-sensitive learning* (Elkan, 2001).

Recently, long-tail learning has received renewed interest in the context of neural networks. Two active strands of work involve post-hoc normalisation of the classification weights (Zhang et al., 2019; Kim & Kim, 2019; Kang et al., 2020; Ye et al., 2020), and modification of the underlying loss to account for varying class penalties (Zhang et al., 2017; Cui et al., 2019; Cao et al., 2019; Tan et al., 2020). Each of these strands are intuitive, and have proven empirically successful. However, they are not without limitation: e.g., weight normalisation crucially relies on the weight norms being smaller for rare classes; however, this assumption is sensitive to the choice of optimiser (see §2.1). On the other hand, loss modification sacrifices the *consistency* that underpins the canonical softmax cross-entropy (see §5.1). Consequently, such techniques may prove suboptimal even in simple settings (see §6.1).

In this paper, we establish a statistical framework for long-tail learning that offers a unified view of post-hoc normalisation and loss modification techniques, while overcoming their limitations. Our framework revisits the classic idea of *logit adjustment* based on label frequencies (Provost, 2000; Zhou & Liu, 2006; Collell et al., 2016), which encourages a large *relative margin* between a pair of rare positive and dominant negative labels. Such adjustment can be achieved by shifting the learned logits post-hoc, or augmenting the softmax cross-entropy with a *pairwise label margin* (cf. (11)). While similar in nature to recent techniques, our logit adjustment approaches additionally have a firm statistical grounding: they are *Fisher consistent* for minimising the *balanced error* (cf. (2)), a common metric in long-tail settings which averages the per-class errors. This statistical grounding translates into strong empirical performance on four real-world datasets with long-tailed label distributions.

In summary, our contributions are:

(i) we establish a statistical framework for long-tail learning (§3) based on *logit adjustment* that provides a unified view of post-hoc correction and loss modification

Table 1: Comparison of approaches to long-tail learning. Weight normalisation re-scales the classification weights; by contrast, we *add* per-label offsets to the logits. Margin approaches uniformly increase the margin between a rare positive and all negatives (Cao et al., 2019), or decrease the margin between all positives and a rare negative (Tan et al., 2020) to prevent rare labels' gradient suppression. By contrast, we increase the margin between a *rare* positive and a *dominant* negative.

| Method | Procedure | Consistent? | Reference |
|---|---|---|---|
| Weight normalisation | Post-hoc weight scaling | × | (Kang et al., 2020) |
| Adaptive margin | Softmax with rare +ve upweighting | × | (Cao et al., 2019) |
| Equalised margin | Softmax with rare -ve downweighting | × | (Tan et al., 2020) |
| Logit-adjusted threshold | Post-hoc logit translation | ✓ | This paper (cf. (9)) |
| Logit-adjusted loss | Softmax with logit translation | ✓ | This paper (cf. (10)) |

(ii) we present two realisations of logit adjustment, applied either post-hoc (§4.1) or during training (§5.1); unlike recent proposals (Table 1), these are consistent for minimising the balanced error

(iii) we confirm the efficacy of the proposed logit adjustment techniques compared to several baselines on four real-world datasets with long-tailed label distributions (§6).

## 2 PROBLEM SETUP AND RELATED WORK

Consider a multiclass classification problem with instances $\mathcal{X}$ and labels $\mathcal{Y} = [L] \doteq \{1, 2, \ldots, L\}$. Given a sample $S = \{(x_n, y_n)\}_{n=1}^N \sim \mathbb{P}^N$ for unknown distribution $\mathbb{P}$ over $\mathcal{X} \times \mathcal{Y}$, our goal is to learn a scorer $f \colon \mathcal{X} \to \mathbb{R}^L$ that minimises the misclassification error $\mathbb{P}_{x,y}\big(y \notin \operatorname{argmax}_{y' \in \mathcal{Y}} f_{y'}(x)\big)$. Typically, one minimises a surrogate loss $\ell \colon \mathcal{Y} \times \mathbb{R}^L \to \mathbb{R}$ such as the softmax cross-entropy,

$$\ell(y, f(x)) = \log \Big[ \sum_{y' \in [L]} e^{f_{y'}(x)} \Big] - f_y(x) = \log \Big[ 1 + \sum_{y' \neq y} e^{f_{y'}(x) - f_y(x)} \Big]. \tag{1}$$

We may view the resulting softmax probabilities $p_y(x) \propto e^{f_y(x)}$ as estimates of $\mathbb{P}(y \mid x)$.

The setting of *learning under class imbalance* or *long-tail learning* is where the distribution $\mathbb{P}(y)$ is highly skewed, so that many rare (or "tail") labels have a low probability of occurrence. Here, the misclassification error is not a suitable measure of performance: a trivial predictor which classifies every instance to the majority label will attain a low misclassification error. To cope with this, a natural alternative is the balanced error (Chan & Stolfo, 1998; Brodersen et al., 2010; Menon et al., 2013), which averages each of the per-class error rates: under a uniform label distribution

$$\mathrm{BER}(f) \doteq \frac{1}{L} \sum_{y \in [L]} \mathbb{P}_{x|y}\big(y \notin \operatorname{argmax}_{y' \in \mathcal{Y}} f_{y'}(x)\big). \tag{2}$$

This can be seen as implicitly using a *balanced* class-probability function $\mathbb{P}^{\mathrm{bal}}(y \mid x) \propto \frac{1}{L} \cdot \mathbb{P}(x \mid y)$, as opposed to the native $\mathbb{P}(y \mid x) \propto \mathbb{P}(y) \cdot \mathbb{P}(x \mid y)$ that is employed in the misclassification error.[1]

Broadly, extant approaches to coping with class imbalance modify:

(i) the *inputs* to a model, for example by over- or under-sampling (Kubat & Matwin, 1997; Chawla et al., 2002; Wallace et al., 2011; Mikolov et al., 2013; Mahajan et al., 2018; Yin et al., 2018);

(ii) the *outputs* of a model, for example by post-hoc correction of the decision threshold (Fawcett & Provost, 1996; Collell et al., 2016) or weights (Kim & Kim, 2019; Kang et al., 2020); or

(iii) the *training procedure* of a model, for example by modifying the loss function (Zhang et al., 2017; Cui et al., 2019; Cao et al., 2019; Tan et al., 2020; Jamal et al., 2020).

One may easily combine approaches from the first stream with those from the latter two. Consequently, we focus on the latter two in this work, and describe some representative recent examples from each.

**Post-hoc weight normalisation**. Suppose $f_y(x) = w_y^\top \Phi(x)$ for classification weights $w_y \in \mathbb{R}^D$ and representations $\Phi \colon \mathcal{X} \to \mathbb{R}^D$, as learned by a neural network. (We may add per-label bias terms

---

[1]Both the misclassification and balanced error compare the top-1 predicted versus true label. One may analogously define a balanced *top-k error* (Lapin et al., 2018), which may be useful in retrieval settings.

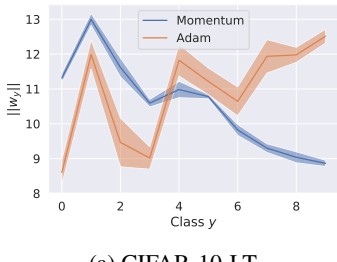

(a) CIFAR-10-LT.

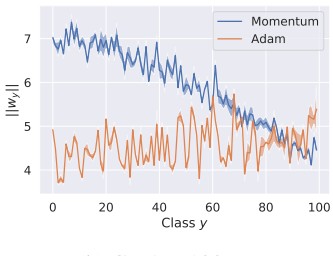

(b) CIFAR-100-LT.

Figure 1: Mean and standard deviation of per-class weight norms $\|w_y\|_2$ over 5 runs for a ResNet-32 under momentum and Adam optimisers. We use long-tailed ("LT") versions of CIFAR-10 and CIFAR-100, and sort classes in descending order of frequency; the first class is $100\times$ more likely to appear than the last class (see §6.2). Both optimisers yield comparable balanced error. However, the weight norms have incompatible trends: under momentum, the norms are strongly correlated with class frequency, while with Adam, the norms are *anti-correlated* or *independent* of the class frequency. Consequently, weight normalisation under Adam is ineffective for combatting class imbalance.

to $f_y$ by adding a constant feature to $\Phi$.) A fruitful avenue of exploration involves decoupling of representation and classifier learning (Zhang et al., 2019). Concretely, we first learn $\{w_y, \Phi\}$ via standard training on the long-tailed training sample $S$, and then for $x \in \mathcal{X}$ predict the label

$$\operatorname{argmax}_{y \in [L]} w_y^\top \Phi(x)/\nu_y^\tau = \operatorname{argmax}_{y \in [L]} f_y(x)/\nu_y^\tau, \tag{3}$$

for $\tau > 0$, where $\nu_y = \mathbb{P}(y)$ in Kim & Kim (2019); Ye et al. (2020) and $\nu_y = \|w_y\|_2$ in Kang et al. (2020). Intuitively, either choice of $\nu_y$ upweights the contribution of rare labels through *weight normalisation*. The choice $\nu_y = \|w_y\|_2$ is motivated by the observations that $\|w_y\|_2$ tends to correlate with $\mathbb{P}(y)$. Further to the above, one may enforce $\|w_y\|_2 = 1$ during training (Kim & Kim, 2019).

**Loss modification**. A classic means of coping with class imbalance is to *balance* the loss, wherein $\ell(y, f(x))$ is weighted by $\mathbb{P}(y)^{-1}$ (Xie & Manski, 1989; Morik et al., 1999): e.g., applied to (1),

$$\ell(y, f(x)) = \frac{1}{\mathbb{P}(y)} \cdot \log\left[1 + \sum_{y' \neq y} e^{f_{y'}(x) - f_y(x)}\right]. \tag{4}$$

While intuitive, balancing has minimal effect in separable settings: solutions that achieve zero training loss will necessarily remain optimal even under weighting (Byrd & Lipton, 2019). Intuitively, one would instead like to shift the separator closer to a dominant class. Li et al. (2002); Wu et al. (2008); Masnadi-Shirazi & Vasconcelos (2010) thus proposed to add *per-class margins* into the hinge loss. Cao et al. (2019) similarly proposed to add a per-class margin into the softmax cross-entropy:

$$\ell(y, f(x)) = \log\left[1 + \sum_{y' \neq y} e^{\delta_y} \cdot e^{f_{y'}(x) - f_y(x)}\right], \tag{5}$$

where $\delta_y \propto \mathbb{P}(y)^{-1/4}$. This upweights rare "positive" $y$ to encourage a larger gap $f_y(x) - f_{y'}(x)$, i.e., the margin between $y$ and any "negative" $y' \neq y$. Separately, Tan et al. (2020) proposed

$$\ell(y, f(x)) = \log\left[1 + \sum_{y' \neq y} e^{\delta_{y'}} \cdot e^{f_{y'}(x) - f_y(x)}\right], \tag{6}$$

where $\delta_{y'} \leq 0$ is a non-decreasing transform of $\mathbb{P}(y')$. Note that in the original softmax cross-entropy with $\delta_{y'} = 0$, a rare label often receives a strong *inhibitory* gradient signal as it disproportionately appear as a negative for dominant labels. This can be modulated by letting $\delta_{y'} \ll 0$.

## 2.1 LIMITATIONS OF EXISTING APPROACHES

Each of the above methods are intuitive, and have shown strong empirical performance. However, a closer analysis identifies some subtle limitations.

*Limitations of weight normalisation*. Post-hoc weight normalisation with $\nu_y = \|w_y\|_2$ per Kang et al. (2020) is motivated by the observation that the weight norm $\|w_y\|_2$ tends to correlate with $\mathbb{P}(y)$. However, this assumption is highly dependent on the choice of optimiser, as Figure 1 illustrates: for

ResNet-32 models trained on long-tailed versions of CIFAR-10 and CIFAR-100, when using the Adam optimiser, the norms are either *anti*-correlated or *independent* of $\mathbb{P}(y)$. Weight normalisation thus cannot achieve the desired effect of boosting rare labels' scores. One may hope to side-step this by simply using $\nu_y = \mathbb{P}(y)$; unfortunately, this choice has more subtle limitations (see §4.2).

*Limitations of loss modification.* Enforcing a per-label margin per (5) and (6) is intuitive, as it allows for shifting the decision boundary away from rare classes. However, when doing so, it is important to ensure *Fisher consistency* (Lin, 2004) (or *classification calibration* (Bartlett et al., 2006)) of the resulting loss for the balanced error. That is, the minimiser of the expected loss (equally, the empirical risk in the infinite sample limit) should result in a minimal balanced error. Unfortunately, both (5) and (6) are *not* consistent in this sense, even for binary problems; see §5.1, §6.1 for details.

# 3  LOGIT ADJUSTMENT FOR LONG-TAIL LEARNING: A STATISTICAL VIEW

The above suggests that there is scope for improving performance on long-tail problems, both in terms of post-hoc correction and loss modification. We now show how a statistical perspective suggests simple procedures of each type, both of which overcome the limitations discussed above.

Recall that our goal is to minimise the balanced error (2). A classical result is that the *best possible* or *Bayes-optimal* scorer for this problem, i.e., $f^* \in \mathrm{argmin}_{f\colon \mathcal{X} \to \mathbb{R}^L} \mathrm{BER}(f)$, satisfies the following (Menon et al., 2013), (Koyejo et al., 2014, Corollary 4), (Collell et al., 2016, Theorem 1):

$$\mathrm{argmax}_{y\in[L]}\, f_y^*(x) = \mathrm{argmax}_{y\in[L]}\, \mathbb{P}^{\mathrm{bal}}(y \mid x) = \mathrm{argmax}_{y\in[L]}\, \mathbb{P}(x \mid y), \qquad (7)$$

where $\mathbb{P}^{\mathrm{bal}}$ is the balanced class-probability as per §2. In words, the Bayes-optimal prediction is the label under which the given instance $x \in \mathcal{X}$ is most likely. Consequently, for fixed class-conditionals $\mathbb{P}(x \mid y)$, varying the *class priors* $\mathbb{P}(y)$ arbitrarily will not affect the optimal scorers. This is intuitively desirable: the balanced error is agnostic to the level of imbalance in the label distribution.

To further probe (7), suppose the underlying class-probabilities $\mathbb{P}(y \mid x) \propto \exp(s_y^*(x))$, for scorer $s^*\colon \mathcal{X} \to \mathbb{R}^L$. Since by definition $\mathbb{P}^{\mathrm{bal}}(y \mid x) \propto \mathbb{P}(y \mid x)/\mathbb{P}(y)$, (7) becomes

$$\mathrm{argmax}_{y\in[L]}\, \mathbb{P}^{\mathrm{bal}}(y \mid x) = \mathrm{argmax}_{y\in[L]}\, \exp(s_y^*(x))/\mathbb{P}(y) = \mathrm{argmax}_{y\in[L]}\, s_y^*(x) - \ln\mathbb{P}(y), \quad (8)$$

i.e., we translate the optimal logits based on the class priors. Equation 8 provides the ideal predictions for optimising the balanced error, which necessarily rely on unknown quantities $s_y^*(x), \mathbb{P}(y)$ that depend on the underlying distribution. Nonetheless, we may seek to *approximate* these quantities based on our training sample. Concretely, (8) suggests two means of optimising for the balanced error:

(i) train a model to estimate the standard $\mathbb{P}(y \mid x)$ (e.g., by minimising the standard softmax-cross entropy on the long-tailed data), and then explicitly modify its logits post-hoc as per (8)

(ii) train a model to estimate the balanced $\mathbb{P}^{\mathrm{bal}}(y \mid x)$, whose logits are implicitly modified as per (8).

Such *logit adjustment* techniques — which have been a classic approach to class-imbalance (Provost, 2000) — neatly align with the post-hoc and loss modification streams discussed in §2. However, unlike most previous techniques from these streams, logit adjustment is endowed with a clear statistical grounding: by construction, the optimal solution under such adjustment coincides with the Bayes-optimal solution (7) for the balanced error, i.e., it is *Fisher consistent* for minimising the balanced error. We now study each of the techniques (i) and (ii) in turn.

# 4  POST-HOC LOGIT ADJUSTMENT

We now propose a post-hoc logit adjustment scheme for a classifier trained on long-tailed data. We further show this has subtle advantages over recent weight normalisation schemes.

## 4.1  THE POST-HOC LOGIT ADJUSTMENT PROCEDURE

When employing the softmax cross-entropy to train a neural network, we aim to approximate the underlying $\mathbb{P}(y \mid x)$ with $p_y(x) \propto \exp(f_y(x))$ for logits $f_y(x) = w_y^\top \Phi(x)$. Given learned $\{w, \Phi\}$,

one typically predicts the label $\text{argmax}_{y \in [L]} f_y(x)$, i.e., the most likely label under the model's $\mathbb{P}(y \mid x)$. In *post-hoc logit adjustment*, we propose to instead predict, for suitable $\tau > 0$:

$$\text{argmax}_{y \in [L]} \exp(w_y^\top \Phi(x))/\pi_y^\tau = \text{argmax}_{y \in [L]} f_y(x) - \tau \cdot \log \pi_y, \qquad (9)$$

where $\pi \in \Delta_L$ (for simplex $\Delta$) are estimates of the class priors $\mathbb{P}(y)$, e.g., the empirical frequencies on the training sample $S$. Effectively, (9) adds a label-dependent offset to each of the logits.

When $\tau = 1$, this can be seen as applying (8) with a plugin estimate of $\mathbb{P}(y \mid x)$, i.e., $p_y(x) \propto \exp(w_y^\top \Phi(x))$. When $\tau \neq 1$, this can be seen as applying (8) to *temperature scaled* estimates $\bar{p}_y(x) \propto \exp(\tau^{-1} \cdot w_y^\top \Phi(x))$. To unpack this, recall that (8) justifies post-hoc logit thresholding given access to the true probabilities $\mathbb{P}(y \mid x)$. In practice, high-capacity neural networks often produce uncalibrated estimates of these probabilities (Guo et al., 2017). Temperature scaling is a means to calibrate the estimates, and is routinely employed for distillation (Hinton et al., 2015).

One may treat $\tau$ as a tuning parameter to be chosen based on holdout calibration, e.g., the expected calibration error (Murphy & Winkler, 1987; Guo et al., 2017), probabilistic sharpness (Gneiting et al., 2007; Kuleshov et al., 2018), or a proper scoring rule such as the log-loss or squared error (Gneiting & Raftery, 2007). One may alternately fix $\tau = 1$ and aim to learn inherently calibrated probabilities, e.g., via label smoothing (Szegedy et al., 2016; Müller et al., 2019).

Post-hoc logit adjustment with $\tau = 1$ is not a new idea in the classical label imbalance literature (Fawcett & Provost, 1996; Provost, 2000; Maloof, 2003; Zhou & Liu, 2006; Collell et al., 2016); however, it has had limited exploration in the recent long-tail learning literature. Further, the case $\tau \neq 1$ is important in practical usage of neural networks, owing to their typical lack of probabilistic calibration (Guo et al., 2017). Interestingly, post-hoc logit adjustment also has an important advantage over recently proposed weight normalisation techniques, as we now discuss.

### 4.2 COMPARISON TO POST-HOC WEIGHT NORMALISAITON

Recall that weight normalisation involves learning logits $f_y(x) = w_y^\top \Phi(x)$, and then post-hoc normalising the weights via $w_y/\nu_y^\tau$ for $\tau > 0$. We demonstrated in §2 that using $\nu_y = \|w_y\|_2$ may be ineffective when using adaptive optimisers. However, even with $\nu_y = \pi_y$, there is a subtle contrast to post-hoc logit adjustment: while the former performs a *multiplicative* update to the logits, the latter performs an *additive* update. The two techniques may thus yield different orderings over labels, since

$$w_1^\top \Phi(x)/\pi_1 < w_2^\top \Phi(x)/\pi_2 \;\;\overset{\Rightarrow}{\underset{\Leftarrow}{\neq}}\;\; \exp(w_1^\top \Phi(x))/\pi_1 < \exp(w_2^\top \Phi(x))/\pi_2.$$

Observe that if a rare label $y$ has a *negative* score $w_y^\top \Phi(x) < 0$, and there is another label with a positive score, then it is *impossible* for the weight normalisation to give $y$ the highest score. By contrast, under logit adjustment, $w_y^\top \Phi(x) - \ln \pi_y$ will be lower for dominant classes, regardless of the original sign. Weight normalisation is thus *not* consistent for the balanced error, unlike logit adjustment.

## 5 THE LOGIT ADJUSTED SOFTMAX CROSS-ENTROPY

We now show how to directly encode logit adjustment into the softmax cross-entropy. The resulting approach has an intuitive relation to existing loss modification techniques.

### 5.1 THE LOGIT ADJUSTED LOSS

From §3, the second approach to optimising for the balanced error is to directly model $\mathbb{P}^{\text{bal}}(y \mid x) \propto \mathbb{P}(y \mid x)/\mathbb{P}(y)$. To do so, consider the following *logit adjusted softmax cross-entropy loss* for $\tau > 0$:

$$\ell(y, f(x)) = -\log \frac{e^{f_y(x) + \tau \cdot \log \pi_y}}{\sum_{y' \in [L]} e^{f_{y'}(x) + \tau \cdot \log \pi_{y'}}} = \log \left[ 1 + \sum_{y' \neq y} \left( \frac{\pi_{y'}}{\pi_y} \right)^\tau \cdot e^{f_{y'}(x) - f_y(x)} \right]. \quad (10)$$

Given a scorer that minimises the above, we now predict $\text{argmax}_{y \in [L]} f_y(x)$ as usual.

Compared to the standard softmax cross-entropy (1), the above applies a *label-dependent offset* to each logit. Compared to (9), we *directly* enforce the class prior offset while learning the logits,

rather than doing this post-hoc. The two approaches have a deeper connection: observe that (10) is equivalent to using a scorer of the form $g_y(x) = f_y(x) + \tau \cdot \log \pi_y$, with $\text{argmax}_{y \in [L]} f_y(x) = \text{argmax}_{y \in [L]} g_y(x) - \tau \cdot \log \pi_y$. Consequently, one can equivalently view learning with this loss as learning a standard scorer $g(x)$, and post-hoc adjusting its logits. For non-convex objectives, as encountered in neural networks, the bias endowed by adding $\tau \cdot \log \pi_y$ to the logits is likely to result in a different local minima, typically with improved performance.

## 5.2 COMPARISON TO LOSS MODIFICATION TECHNIQUES

For more insight into the logit adjusted loss, consider the following *pairwise margin loss*

$$\ell(y, f(x)) = \alpha_y \cdot \log \left[ 1 + \sum_{y' \neq y} e^{\Delta_{yy'}} \cdot e^{(f_{y'}(x) - f_y(x))} \right], \tag{11}$$

for label weights $\alpha_y > 0$, and *pairwise label margins* $\Delta_{yy'}$ representing the desired gap between scores for $y$ and $y'$. For $\tau = 1$, our logit adjusted loss (10) corresponds to (11) with $\alpha_y = 1$ and $\Delta_{yy'} = \log \left( \frac{\pi_{y'}}{\pi_y} \right)$. This demands a larger margin between *rare* positive ($\pi_y \sim 0$) and *dominant* negative ($\pi_{y'} \sim 1$) labels, so that scores for dominant classes do not overwhelm those for rare ones.

Existing loss modification techniques can be viewed as special cases of (11). For example, $\alpha_y = 1/\pi_y$ and $\Delta_{yy'} = 0$ yields the balanced loss (4). When $\alpha_y = 1$, the choice $\Delta_{yy'} = \pi_y^{-1/4}$ yields (5). Finally, $\Delta_{yy'} = \log F(\pi_{y'})$ yields (6), where $F : [0, 1] \to (0, 1]$ is some non-decreasing function. These losses thus either consider the frequency of the positive $y$ or negative $y'$, but not *both*. Remarkably, the specific choice that leads to our loss in (10) has a firm statistical grounding: it ensures Fisher consistency (in the sense of, e.g., Bartlett et al. (2006)) for the balanced error. Proofs for all results are in the supplementary.

**Theorem 1.** *For any $\delta \in \mathbb{R}_+^L$, the pairwise loss in (11) is Fisher consistent with weights and margins*

$$\alpha_y = \delta_y / \mathbb{P}(y) \qquad \Delta_{yy'} = \log \left( \delta_{y'} / \delta_y \right).$$

Letting $\delta_y = \pi_y$, we immediately deduce that the logit-adjusted loss of (10) is consistent, *provided* our $\pi_y$ is a consistent estimate of $\mathbb{P}(y)$. Similarly, $\delta_y = 1$ recovers the classic result that the balanced loss is consistent. While Theorem 1 only provides a sufficient condition in multi-class setting, one can provide a necessary and sufficient condition that rules out other choices of $\Delta$ in the binary case.

**Theorem 2.** *Suppose $\mathcal{Y} = \{\pm 1\}$. Let $\delta_y \doteq \Delta_{y,-y}$, and $\sigma(z) = (1 + \exp(z))^{-1}$. Then, the pairwise margin loss in (11) is Fisher consistent for the balanced error iff*

$$\frac{\alpha_{+1}}{\alpha_{-1}} \cdot \frac{\sigma(\delta_{+1})}{\sigma(\delta_{-1})} = \frac{1 - \mathbb{P}(y = +1)}{\mathbb{P}(y = +1)}.$$

## 5.3 DISCUSSION AND EXTENSIONS

Our pairwise margin loss in (11) subsumes several existing loss-correction approaches in the literature. Further, it suggests the exploration of new choices of $\Delta$. For example, we shall see the efficacy of combining the $\Delta$ implicit in the adaptive loss of Cao et al. (2019) with our proposed $\Delta$. One may also generalise the formulation in Theorem 1, and employ $\Delta_{yy'} = \tau_1 \cdot \log \pi_y - \tau_2 \cdot \log \pi_{y'}$, where $\tau_1, \tau_2 > 0$. This interpolates between our loss ($\tau_1 = \tau_2$) and a version of the equalised loss ($\tau_1 = 0$).

For $\tau = -1$, a similar loss to (10) has been considered in the context of *negative sampling* for scalability (Yi et al., 2019): here, one samples a subset of negatives based on $\pi$, and corrects the logits to obtain an unbiased estimate of the loss based on all negatives (Bengio & Senecal, 2008). Losses of the general form (11) have also been explored for structured prediction (Pletscher et al., 2010).

Cao et al. (2019, Theorem 2) provides a rigorous generalisation bound for the adaptive margin loss under the assumption of separable data with binary labels. The inconsistency of the loss with respect to the balanced error concerns the more general scenario of non-separable multiclass data, which may occur, e.g., owing to label noise or limited model capacity. We shall subsequently demonstrate that encouraging consistency is not merely of theoretical interest, and can lead to gains in practice.

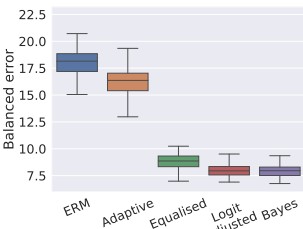 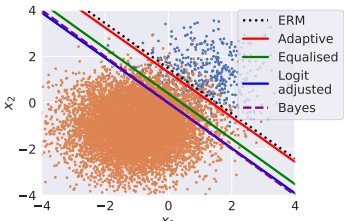 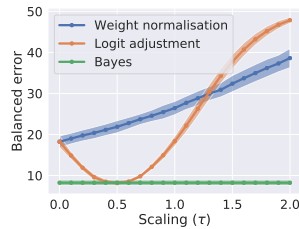

Figure 2: Results on synthetic binary classification problem. Our logit adjusted loss tracks the Bayes-optimal solution and separator (left & middle panel). Post-hoc logit adjustment matches the Bayes performance with suitable scaling (right panel); however, *any* weight normalisation fails.

## 6 EXPERIMENTAL RESULTS

We now present experiments confirming our main claims: (i) on simple binary problems, existing weight normalisation and loss modification techniques may not converge to the optimal solution (§6.1); (ii) on real-world datasets, our post-hoc logit adjustment generally outperforms weight normalisation, and one can obtain further gains via our logit adjusted softmax cross-entropy (§6.2).

### 6.1 RESULTS ON SYNTHETIC DATASET

We begin with a binary classification task, wherein samples from class $y \in \{\pm 1\}$ are drawn from a 2D Gaussian with isotropic covariance and means $\mu_y = y \cdot (+1, +1)$. We introduce class imbalance by setting $\mathbb{P}(y = +1) = 5\%$. The Bayes-optimal classifier for the balanced error is (see Appendix F)

$$f^*(x) = +1 \iff \mathbb{P}(x \mid y = +1) > \mathbb{P}(x \mid y = -1) \iff (\mu_1 - \mu_{-1})^\top x > 0, \quad (12)$$

i.e., it is a linear separator passing through the origin. We compare this separator against those found by several margin losses based on (11): standard ERM ($\Delta_{yy'} = 0$), the adaptive loss (Cao et al., 2019) ($\Delta_{yy'} = \pi_y^{-1/4}$), an instantiation of the equalised loss (Tan et al., 2020) ($\Delta_{yy'} = \log \pi_{y'}$), and our logit adjusted loss ($\Delta_{yy'} = \log \frac{\pi_{y'}}{\pi_y}$). For each loss, we train an affine classifier on a sample of $10,000$ instances, and evaluate the balanced error on a test set of $10,000$ samples over $100$ independent trials.

Figure 2 confirms that the logit adjusted margin loss attains a balanced error close to that of the Bayes-optimal, which is visually reflected by its learned separator closely matching that in (12). This is in line with our claim of the logit adjusted margin loss being consistent for the balanced error, unlike other approaches. Figure 2 also compares post-hoc weight normalisation and logit adjustment for varying scaling parameter $\tau$ (c.f. (3), (9)). Logit adjustment is seen to approach the performance of the Bayes predictor; *any* weight normalisation is however seen to hamper performance. This verifies the consistency of logit adjustment, and inconsistency of weight normalisation (§4.2).

### 6.2 RESULTS ON REAL-WORLD DATASETS

We present results on the CIFAR-10, CIFAR-100, ImageNet and iNaturalist 2018 datasets. Following prior work, we create "long-tailed versions" of the CIFAR datasets by suitably downsampling examples per label following the EXP profile of Cui et al. (2019); Cao et al. (2019) with imbalance ratio $\rho = \max_y \mathbb{P}(y)/\min_y \mathbb{P}(y) = 100$. Similarly, we use the long-tailed version of ImageNet produced by Liu et al. (2019). We employ a ResNet-32 for CIFAR, and a ResNet-50 for ImageNet and iNaturalist. All models are trained using SGD with momentum; see Appendix C for more details. See also Appendix D.3 for results on CIFAR under the STEP profile also considered in the literature.

**Baselines**. We consider several representative baselines: (i) empirical risk minimisation (ERM) on the long-tailed data, (ii) post-hoc weight normalisation (Kang et al., 2020) per (3) (using $\nu_y = \|w_y\|_2$) applied to ERM, (iii) the class-balanced loss of Cui et al. (2019), (iv) the adaptive margin loss (Cao et al., 2019) per (5), including with the "deferred reweighting" (DRW) training scheme, and (v) the equalised loss (Tan et al., 2020) per (6), with $\delta_{y'} = F(\pi_{y'})$ for the threshold-based $F$ of Tan et al. (2020). Where possible, we report numbers for the baselines from the respective papers.

Table 2: Test set balanced error (averaged over 5 trials) on real-world datasets. Here, [†], [⋆], [‡] are numbers for "LDAM + SGD" and "LDAM + DRW" from Cao et al. (2019, Table 2, 3); "$\tau$-normalised" from Kang et al. (2020, Table 3, 7); and "Class-Balanced" from Cui et al. (2019, Table 2, 3). Here, $\tau = \tau^*$ refers to using the best possible tuning parameter $\tau$; see Figure 3 for results on various $\tau$. Highlighted cells denote the best performing method for a given dataset.

| Method | CIFAR-10-LT | CIFAR-100-LT | ImageNet-LT | iNaturalist |
|---|---|---|---|---|
| ERM | 27.16 | 61.64 | 53.11 | 38.66 |
| Weight normalisation ($\tau = 1$) (Kang et al., 2020) | 24.02 | 58.89 | 52.00 | 48.05 |
| Weight normalisation ($\tau = \tau^*$) (Kang et al., 2020) | 21.50 | 58.76 | 49.37 | 34.40[⋆] |
| Class-balanced (Cui et al., 2019) | 25.43[‡] | 60.40[‡] | 53.21 | 35.84[‡] |
| Adaptive (Cao et al., 2019) | 26.65[†] | 60.40[†] | 52.15 | 33.31 |
| Adaptive + DRW (Cao et al., 2019) | 22.97[†] | 57.96[†] | 49.85 | 32.00[†] |
| Equalised (Tan et al., 2020) | 26.02 | 57.26 | 54.02 | 38.37 |
| Logit adjustment post-hoc ($\tau = 1$) | 22.60 | 58.24 | 49.66 | 33.98 |
| Logit adjustment loss ($\tau = 1$) | 22.33 | 56.11 | 48.89 | 33.64 |
| Logit adjustment plus adaptive loss ($\tau = 1$) | 22.42 | 55.92 | 51.25 | 31.56 |

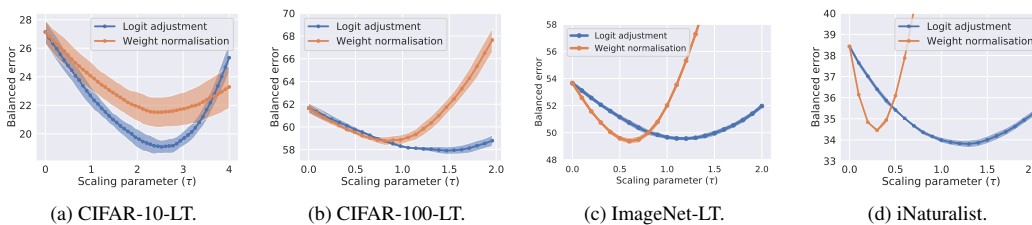

(a) CIFAR-10-LT.     (b) CIFAR-100-LT.     (c) ImageNet-LT.     (d) iNaturalist.

Figure 3: Comparison of balanced error for post-hoc correction techniques when varying scaling parameter $\tau$ (c.f. (3), (9)). Post-hoc logit adjustment consistently outperforms weight normalisation.

We compare the above methods against our proposed post-hoc logit adjustment (9), and logit adjusted loss (10). For post-hoc logit adjustment, we fix the scalar $\tau = 1$ for our basic results; we analyse the effect of tuning this in Figure 3. We additionally evaluate a combination of our logit adjusted softmax cross-entropy with the adaptive margin of Cao et al. (2019); this uses (11) with $\Delta_{yy'} = \log(\pi_{y'}/\pi_y) + \pi_y^{-1/4}$. We do not perform *any* further tuning of our techniques. For all methods, we report the balanced error on the test set. (Note that since the test sets are all balanced for these benchmarks, the balanced error is equivalent to the misclassification error.) For all methods, we pre-compute $\pi$ as the empirical label frequency on the entire training set.

**Results and analysis**. Table 2 demonstrates that our proposed logit adjustment techniques consistently outperform existing methods. Indeed, weight normalisation with $\tau = 1$ is generally improved significantly by post-hoc logit adjustment (e.g., 8% relative reduction on CIFAR-10). Similarly, loss correction techniques are generally outperformed by our logit adjusted softmax cross-entropy (e.g., 6% relative reduction on iNaturalist). Cao et al. (2019) observed that their loss benefits from a deferred reweighting scheme (DRW), wherein class-weighting is applied after a fixed number of epochs. Table 2 indicates this is consistently outperformed by suitable variants of logit adjustment.

Table 2 only reports the results for logit adjustment with scalar $\tau = 1$. In practice, tuning $\tau$ can significantly improve performance further. Figure 3 studies the effect of tuning $\tau$ for post-hoc weight normalisation (using $\nu_y = \|w_y\|_2$) and post-hoc logit adjustment. Even without *any* scaling, post-hoc logit adjustment generally offers superior performance to the best result from weight normalisation (cf. Table 2); with scaling, this is further improved. In practice, one may choose $\tau$ via cross-validation against the balanced error on the training set. For example, on CIFAR-10-LT, we estimate $\tau^* = 2.6$ for post-hoc logit adjustment, for which the resulting balanced test error of **18.73**% is superior to that of weight normalisation for any $\tau$.

To better understand the gains, Figure 4 reports errors on a per-group basis, where following Kang et al. (2020) we construct three groups of classes — "Many", "Medium", and "Few" — comprising

Figure 4: Comparison of per-group errors for loss modification techniques. We construct three groups of classes: "Many", comprising those with at least 100 training examples; "Medium", comprising those with at least 20 and at most 100 training examples; and "Few", comprising those with at most 20 training examples.

those with $\geq 100$, between $(20, 100)$, and $\leq 20$ training examples respectively. Logit adjustment shows consistent gains over the "Medium" and "Few" groups, albeit at some expense in "Many" group performance. See Appendix D.2 for a finer-grained breakdown.

While both logit adjustment techniques perform similarly, there is a slight advantage to the loss function version. Nonetheless, the strong performance of post-hoc logit adjustment corroborates the ability to decouple representation and classifier learning in long-tail settings (Zhang et al., 2019). A reference implementation of our methods is planned for release at:

https://github.com/google-research/google-research/tree/master/logit_adjustment.

## 7 DISCUSSION AND FUTURE WORK

Table 2 shows the advantage of logit adjustment over recent proposals, under standard setups from the literature. Further improvements are possible by fusing complementary ideas, and we remark on a few such options. First, one may use a more complex base architecture; e.g., Kang et al. (2020) found gains by employing a ResNet-152, and training for 200 epochs. Table 3 (Appendix) confirms that logit adjustment similarly benefits from this choice, achieving a balanced error of $\mathbf{30.12}\%$ on iNaturalist, and $\mathbf{28.02}\%$ when combined with the adaptive margin. Second, the DRW training scheme (which applies to any loss) may result in further gains for our techniques. Third, incorporating developments in meta-learning (Wang et al., 2017; Jamal et al., 2020) is also a promising avenue. While further exploring such variants are of empirical interest, we hope to have illustrated the conceptual and empirical value of logit adjustment, and leave this for future work.

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

# Supplementary material for "Long tail learning via logit adjustment"

## A    PROOFS OF RESULTS IN BODY

*Proof of Theorem 1.* Denote $\eta_y(x) = \mathbb{P}(y \mid x)$. Suppose we employ a margin $\Delta_{yy'} = \log \frac{\delta_{y'}}{\delta_y}$. Then, the loss is

$$\ell(y, f(x)) = -\log \frac{\delta_y \cdot e^{f_y(x)}}{\sum_{y' \in [L]} \delta_{y'} \cdot e^{f_{y'}(x)}} = -\log \frac{e^{f_y(x) + \log \delta_y}}{\sum_{y' \in [L]} e^{f_{y'}(x) + \log \delta_{y'}}}.$$

Consequently, under constant weights $\alpha_y = 1$, the Bayes-optimal score will satisfy $f_y^*(x) + \log \delta_y = \log \eta_y(x)$, or $f_y^*(x) = \log \frac{\eta_y(x)}{\delta_y}$.

Now suppose we use generic weights $\alpha \in \mathbb{R}_+^L$. The risk under this loss is

$$\begin{aligned}
\mathbb{E}_{x,y}\left[\ell_\alpha(y, f(x))\right] &= \sum_{y \in [L]} \pi_y \cdot \mathbb{E}_{x|y=y}\left[\ell_\alpha(y, f(x))\right] \\
&= \sum_{y \in [L]} \pi_y \cdot \mathbb{E}_{x|y=y}\left[\ell_\alpha(y, f(x))\right] \\
&= \sum_{y \in [L]} \pi_y \cdot \alpha_y \cdot \mathbb{E}_{x|y=y}\left[\ell(y, f(x))\right] \\
&\propto \sum_{y \in [L]} \bar{\pi}_y \cdot \mathbb{E}_{x|y=y}\left[\ell(y, f(x))\right],
\end{aligned}$$

where $\bar{\pi}_y \propto \pi_y \cdot \alpha_y$. Consequently, learning with the weighted loss is equivalent to learning with the original loss, on a distribution with modified base-rates $\bar{\pi}$. Under such a distribution, we have class-conditional distribution

$$\bar{\eta}_y(x) = \bar{\mathbb{P}}(y \mid x) = \frac{\mathbb{P}(x \mid y) \cdot \bar{\pi}_y}{\bar{\mathbb{P}}(x)} = \eta_y(x) \cdot \frac{\bar{\pi}_y}{\pi_y} \cdot \frac{\mathbb{P}(x)}{\bar{\mathbb{P}}(x)} \propto \eta_y(x) \cdot \alpha_y.$$

Consequently, suppose $\alpha_y = \frac{\delta_y}{\pi_y}$. Then, $f_y^*(x) = \log \frac{\bar{\eta}_y(x)}{\delta_y} = \log \frac{\eta_y(x)}{\pi_y} + C(x)$, where $C(x)$ does not depend on $y$. Consequently, $\mathrm{argmax}_{y \in [L]} f_y^*(x) = \mathrm{argmax}_{y \in [L]} \frac{\eta_y(x)}{\pi_y}$, which is the Bayes-optimal prediction for the balanced error.

In sum, a consistent family can be obtained by choosing any set of constants $\delta_y > 0$ and setting

$$\alpha_y = \frac{\delta_y}{\pi_y}$$

$$\Delta_{yy'} = \log \frac{\delta_{y'}}{\delta_y}.$$

$\square$

*Proof of Theorem 2.* We establish a more general result in Lemma 3 of the next section, which allows for a temperature parameter in the loss. This allows for interpolating between the standard softmax cross-entropy and margin based losses. $\square$

## B    ON THE CONSISTENCY OF BINARY MARGIN-BASED LOSSES

It is instructive to study the pairwise margin loss (11) in the binary case. Endowing the loss with a temperature parameter $\gamma > 0$, we get[2]

$$
\begin{aligned}
\ell(+1, f) &= \frac{\omega_{+1}}{\gamma} \cdot \log(1 + e^{\gamma \cdot \delta_{+1}} \cdot e^{-\gamma \cdot f}) \\
\ell(-1, f) &= \frac{\omega_{-1}}{\gamma} \cdot \log(1 + e^{\gamma \cdot \delta_{-1}} \cdot e^{\gamma \cdot f})
\end{aligned}
\tag{13}
$$

for constants $\omega_{\pm 1}, \gamma > 0$ and $\delta_{\pm 1} \in \mathbb{R}$. Here, we have used $\delta_{+1} = \Delta_{+1,-1}$ and $\delta_{-1} = \Delta_{-1,+1}$ for simplicity. The choice $\omega_{\pm 1} = 1, \delta_{\pm 1} = 0$ recovers the temperature scaled binary logistic loss. Evidently, as $\gamma \to +\infty$, these converge to weighted hinge losses with variable margins, i.e.,

$$
\begin{aligned}
\ell(+1, f) &= \omega_{+1} \cdot [\delta_{+1} - f]_+ \\
\ell(-1, f) &= \omega_{-1} \cdot [\delta_{-1} + f]_+.
\end{aligned}
$$

We study two properties of this family losses. First, under what conditions are the losses Fisher consistent for the balanced error? We shall show that in fact there is a simple condition characterising this. Second, do the losses preserve properness of the original binary logistic loss? We shall show that this is always the case, but that the losses involve fundamentally different approximations.

### B.1    CONSISTENCY OF THE BINARY PAIRWISE MARGIN LOSS

Given a loss $\ell$, its *Bayes optimal* solution is $f^* \in \mathrm{argmin}_{f : \, \mathcal{X} \to \mathbb{R}} \mathbb{E}\left[\ell(y, f(x))\right]$. For consistency with respect to the balanced error in the binary case, we require this optimal solution $f^*$ to satisfy $f^*(x) > 0 \iff \eta(x) > \pi$, where $\eta(x) \doteq \mathbb{P}(y = 1 \mid x)$ and $\pi \doteq \mathbb{P}(y = 1)$ (Menon et al., 2013). This is equivalent to a simple condition on the weights $\omega$ and margins $\delta$ of the pairwise margin loss.

**Lemma 3.** *The losses in* (13) *are consistent for the balanced error iff*

$$
\frac{\omega_{+1}}{\omega_{-1}} \cdot \frac{\sigma(\gamma \cdot \delta_{+1})}{\sigma(\gamma \cdot \delta_{-1})} = \frac{1 - \pi}{\pi},
$$

*where* $\sigma(z) = (1 + \exp(z))^{-1}$.

*Proof of Lemma 3.* Denote $\eta(x) \doteq \mathbb{P}(y = +1 \mid x)$, and $\pi \doteq \mathbb{P}(y = +1)$. From Lemma 4 below, the pairwise margin loss is proper composite with invertible link function $\Psi \colon [0, 1] \to \mathbb{R} \cup \{\pm\infty\}$. Consequently, since by definition the Bayes-optimal score for a proper composite loss is $f^*(x) = \Psi(\eta(x))$ (Reid & Williamson, 2010), to have consistency for the balanced error, from (14), (15), we require

$$
\begin{aligned}
\Psi^{-1}(0) = \pi &\iff \frac{1}{1 - \frac{\ell'(+1, 0)}{\ell'(-1, 0)}} = \pi \\
&\iff 1 - \frac{\ell'(+1, 0)}{\ell'(-1, 0)} = \frac{1}{\pi} \\
&\iff -\frac{\ell'(+1, 0)}{\ell'(-1, 0)} = \frac{1 - \pi}{\pi} \\
&\iff \frac{\omega_{+1}}{\omega_{-1}} \cdot \frac{\sigma(\gamma \cdot \delta_{+1})}{\sigma(\gamma \cdot \delta_{-1})} = \frac{1 - \pi}{\pi}.
\end{aligned}
$$

$\square$

From the above, some admissible parameter choices include:

- $\omega_{+1} = \frac{1}{\pi}, \omega_{-1} = \frac{1}{1 - \pi}, \delta_{\pm 1} = 1$; i.e., the standard weighted loss with a constant margin

---

[2]Compared to the multiclass case, we assume here a scalar score $f \in \mathbb{R}$. This is equivalent to constraining that $\sum_{y \in [L]} f_y = 0$ for the multiclass case.

- $\omega_{\pm 1} = 1$, $\delta_{+1} = \frac{1}{\gamma} \cdot \log \frac{1-\pi}{\pi}$, $\delta_{-1} = \frac{1}{\gamma} \cdot \log \frac{\pi}{1-\pi}$; i.e., the unweighted loss with a margin biased towards the rare class, as per our logit adjustment procedure

The second example above is unusual in that it requires scaling the margin with the temperature; consequently, the margin disappears as $\gamma \to +\infty$. Other combinations are of course possible, but note that one cannot arbitrarily choose parameters and hope for consistency in general. Indeed, some *inadmissible* choices are naïve applications of the margin modification or weighting, e.g.,

- $\omega_{+1} = \frac{1}{\pi}$, $\omega_{-1} = \frac{1}{1-\pi}$, $\delta_{+1} = \frac{1}{\gamma} \cdot \log \frac{1-\pi}{\pi}$, $\delta_{-1} = \frac{1}{\gamma} \cdot \log \frac{\pi}{1-\pi}$; i.e., combining *both* weighting and margin modification
- $\omega_{\pm 1} = 1$, $\delta_{+1} = \frac{1}{\gamma} \cdot (1 - \pi)$, $\delta_{-1} = \frac{1}{\gamma} \cdot \pi$; i.e., specific margin modification

Note further that the choices of Cao et al. (2019); Tan et al. (2020) do not meet the requirements of Lemma 3.

We make two final remarks. First, the above only considers consistency of the result of loss minimisation. For *any* choice of weights and margins, we may apply suitable post-hoc correction to the predictions to account for any bias in the optimal scores. Second, as $\gamma \to +\infty$, any *constant* margins $\delta_{\pm 1} > 0$ will have no effect on the consistency condition, since $\sigma(\gamma \cdot \delta_{\pm 1}) \to 1$. The condition will be wholly determined by the weights $\omega_{\pm 1}$. For example, we may choose $\omega_{+1} = \frac{1}{\pi}$, $\omega_{-1} = \frac{1}{1-\pi}$, $\delta_{+1} = 1$, and $\delta_{-1} = \frac{\pi}{1-\pi}$; the resulting loss will not be consistent for finite $\gamma$, but will become so in the limit $\gamma \to +\infty$.

### B.2 PROPERNESS OF THE PAIRWISE MARGIN LOSS

In the above, we appealed to the pairwise margin loss being proper composite, in the sense of Reid & Williamson (2010). Intuitively, this specifies that the loss has Bayes-optimal score of the form $f^*(x) = \Psi(\eta(x))$, where $\Psi$ is some invertible function, and $\eta(x) = \mathbb{P}(y = 1 \mid x)$. We have the following general result about properness of *any* member of the pairwise margin family.

**Lemma 4.** *The losses in* (13) *are proper composite, with link function*

$$\Psi(p) = \frac{1}{\gamma} \cdot \log \left[ \left( \frac{a \cdot b}{q} - c \right) \pm \sqrt{\left( \frac{a \cdot b}{q} - c \right)^2 + 4 \cdot \frac{a}{q}} \right] - \log 2,$$

*where* $a = \frac{\omega_{+1}}{\omega_{-1}} \cdot \frac{e^{\gamma \cdot \delta_{+1}}}{e^{\gamma \cdot \delta_{-1}}}$, $b = e^{\gamma \cdot \delta_{-1}}$, $c = e^{\gamma \cdot \delta_{+1}}$, *and* $q = \frac{1-p}{p}$.

*Proof of Lemma 4.* The above family of losses is proper composite iff the function

$$f \mapsto \frac{1}{1 - \frac{\ell'(+1,f)}{\ell'(-1,f)}} \tag{14}$$

is invertible (Reid & Williamson, 2010, Corollary 12); the resulting inverse is the link function $\Psi$. We have

$$\ell'(+1, f) = -\omega_{+1} \cdot \frac{e^{\gamma \cdot \delta_{+1}} \cdot e^{-\gamma \cdot f}}{1 + e^{\gamma \cdot \delta_{+1}} \cdot e^{-\gamma \cdot f}}$$
$$\ell'(-1, f) = +\omega_{-1} \cdot \frac{e^{\gamma \cdot \delta_{-1}} \cdot e^{\gamma \cdot f}}{1 + e^{\gamma \cdot \delta_{-1}} \cdot e^{\gamma \cdot f}}. \tag{15}$$

The invertibility of (14) is immediate. To compute the link function $\Psi$, note that

$$p = \frac{1}{1 - \frac{\ell'(+1,f)}{\ell'(-1,f)}} \iff \frac{1}{p} = 1 - \frac{\ell'(+1,f)}{\ell'(-1,f)}$$

$$\iff -\frac{\ell'(+1,f)}{\ell'(-1,f)} = \frac{1-p}{p}$$

$$\iff \frac{\omega_{+1}}{\omega_{-1}} \cdot \frac{e^{\gamma \cdot \delta_{+1}} \cdot e^{-\gamma \cdot f}}{1 + e^{\gamma \cdot \delta_{+1}} \cdot e^{-\gamma \cdot f}} \cdot \frac{1 + e^{\gamma \cdot \delta_{-1}} \cdot e^{\gamma \cdot f}}{e^{\gamma \cdot \delta_{-1}} \cdot e^{\gamma \cdot f}} = \frac{1-p}{p}$$

$$\iff \frac{\omega_{+1}}{\omega_{-1}} \cdot \frac{e^{\gamma \cdot \delta_{+1}}}{e^{\gamma \cdot \delta_{-1}}} \cdot \frac{1}{e^{\gamma \cdot f} + e^{\gamma \cdot \delta_{+1}}} \cdot \frac{1 + e^{\gamma \cdot \delta_{-1}} \cdot e^{\gamma \cdot f}}{e^{\gamma \cdot f}} = \frac{1 - p}{p}$$

$$\iff a \cdot \frac{1 + b \cdot g}{g^2 + c \cdot g} = q,$$

where $a = \frac{\omega_{+1}}{\omega_{-1}} \cdot \frac{e^{\gamma \cdot \delta_{+1}}}{e^{\gamma \cdot \delta_{-1}}}$, $b = e^{\gamma \cdot \delta_{-1}}$, $c = e^{\gamma \cdot \delta_{+1}}$, $g = e^{\gamma \cdot f}$, and $q = \frac{1-p}{p}$. Thus,

$$a \cdot \frac{1 + b \cdot g}{g^2 + c \cdot g} = q \iff \frac{g^2 + c \cdot g}{1 + b \cdot g} = \frac{a}{q}$$

$$\iff g^2 + \left( c - \frac{a \cdot b}{q} \right) \cdot g - \frac{a}{q} = 0$$

$$\iff g = \frac{\left( \frac{a \cdot b}{q} - c \right) \pm \sqrt{\left( \frac{a \cdot b}{q} - c \right)^2 + 4 \cdot \frac{a}{q}}}{2}.$$

$\square$

As a sanity check, suppose $a = b = c = \gamma = 1$. This corresponds to the standard logistic loss. Then,

$$\Psi(p) = \log \frac{\left( \frac{1}{q} - 1 \right) \pm \sqrt{\left( \frac{1}{q} - 1 \right)^2 + 4 \cdot \frac{1}{q}}}{2} = \log \frac{p}{1 - p},$$

which is the standard logit function.

Figure 5 and 6 compares the link functions for a few different settings:

- the balanced loss, where $\omega_{+1} = \frac{1}{\pi}$, $\omega_{-1} = \frac{1}{1-\pi}$, and $\delta_{\pm 1} = 1$
- an unequal margin loss, where $\omega_{\pm 1} = 1$, $\delta_{+1} = \frac{1}{\gamma} \cdot \log \frac{1-\pi}{\pi}$, and $\delta_{-1} = \frac{1}{\gamma} \cdot \log \frac{\pi}{1-\pi}$
- a balanced + margin loss, where $\omega_{+1} = \frac{1}{\pi}$, $\omega_{-1} = \frac{1}{1-\pi}$, $\delta_{+1} = 1$, and $\delta_{-1} = \frac{\pi}{1-\pi}$.

The property $\Psi^{-1}(0) = \pi$ for $\pi = \mathbb{P}(y = 1)$ holds for the first two choices with any $\gamma > 0$, and the third choice as $\gamma \to +\infty$. This indicates the Fisher consistency of these losses for the balanced error. However, the precise way this is achieved is strikingly different in each case. In particular, each loss implicitly involves a fundamentally different link function.

To better understand the effect of parameter choices, Figure 7 illustrates the conditional Bayes risk curves, i.e.,

$$\underline{L}(p) = p \cdot \ell(+1, \Psi(p)) + (1 - p) \cdot \ell(+1, \Psi(p)).$$

We remark here that for the balanced error, this function takes the form $\underline{L}(p) = p \cdot [\![ p < \pi ]\!] + (1 - p) \cdot [\![ p > \pi ]\!]$, i.e., it is a "tent shaped" concave function with a maximum at $p = \pi$.

For ease of comparison, we normalise this curves to have a maximum of 1. Figure 7 shows that simply applying unequal margins does *not* affect the underlying conditional Bayes risk compared to the standard log-loss; thus, the change here is purely in terms of the link function. By contrast, either balancing the loss or applying a combination of weighting and margin modification results in a closer approximation to the conditional Bayes risk curve for the cost-sensitive loss with cost $\pi$.

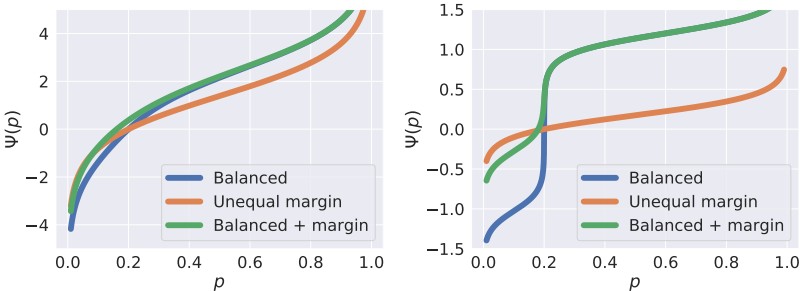

Figure 5: Comparison of link functions for various losses assuming $\pi = 0.2$, with $\gamma = 1$ (left) and $\gamma = 8$ (right). The balanced loss uses $\omega_y = \frac{1}{\pi_y}$. The unequal margin loss uses $\delta_y = \frac{1}{\gamma} \cdot \log \frac{1-\pi}{\pi}$. The balanced + margin loss uses $\delta_{-1} = \frac{\pi}{1-\pi}$, $\delta_{+1} = 1$, $\omega_{+1} = \frac{1}{\pi}$.

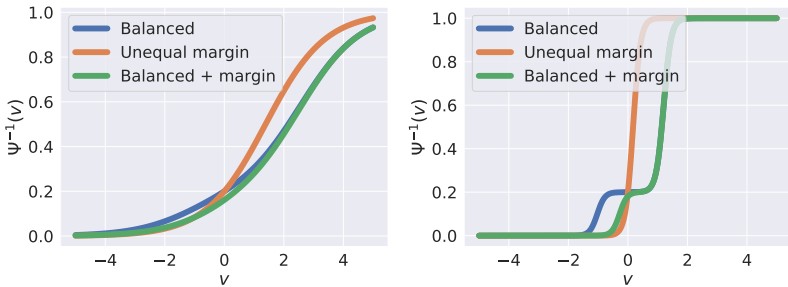

Figure 6: Comparison of link functions for various losses assuming $\pi = 0.2$, with $\gamma = 1$ (left) and $\gamma = 8$ (right). The balanced loss uses $\omega_y = \frac{1}{\pi_y}$. The unequal margin loss uses $\delta_y = \frac{1}{\gamma} \cdot \log \frac{1-\pi_y}{\pi_y}$. The balanced + margin loss uses $\delta_{-1} = \frac{\pi}{1-\pi}$, $\delta_{+1} = 1$, $\omega_{+1} = \frac{1}{\pi}$.

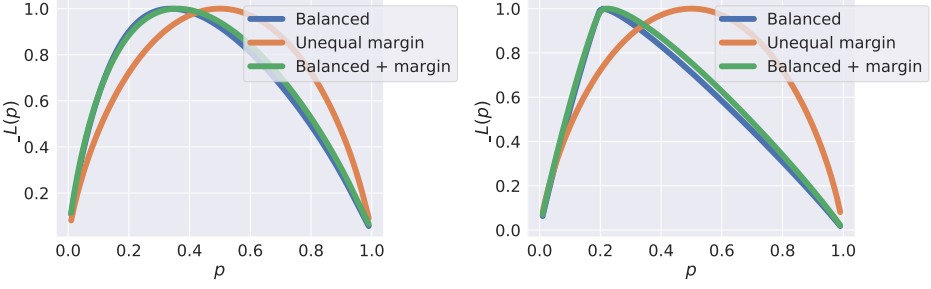

Figure 7: Comparison of conditional Bayes risk functions for various losses assuming $\pi = 0.2$, with $\gamma = 1$ (left) and $\gamma = 8$ (right). The balanced loss uses $\omega_y = \frac{1}{\pi_y}$. The unequal margin loss uses $\delta_y = \frac{1}{\gamma} \cdot \log \frac{1-\pi_y}{\pi_y}$. The first balanced + margin loss uses $\delta_{-1} = \pi$, $\delta_{+1} = 1$, $\omega_{+1} = \frac{1}{\pi}$. The second balanced + margin loss uses $\delta_{-1} = \frac{\pi}{1-\pi}$, $\delta_{+1} = 1$, $\omega_{+1} = \frac{1}{\pi}$.

## C    EXPERIMENTAL SETUP

Intending a fair comparison, we use the same setup for all the methods for each dataset. Unless otherwise specified, all networks are trained with SGD with a momentum value of 0.9, a linear learning rate warm-up in the first 5 epochs to reach the base learning rate, and a weight decay of $10^{-4}$. Other dataset specific details are given below.

**CIFAR-10 and CIFAR-100**: We use a CIFAR ResNet-32 model trained for 120K steps with a batch size of 128. The base learning rate is 0.1, with a linear warmup for the first 2000 steps, and a decay of 0.1 at 60K, 90K, and 110K steps.[3] We also use the standard CIFAR data augmentation procedure used in previous works such as Cao et al. (2019); He et al. (2016), where 4 pixels are padded on each size and a random $32 \times 32$ crop is taken. Images are horizontally flipped with a probability of 0.5.

**ImageNet**: We use a ResNet-50 model trained for 90 epochs with batch size of 512. The base learning rate is 0.4, with cosine learning rate decay and Nesterov momentum. We use a weight decay of $5 \times 10^{-4}$ following Kang et al. (2020). Additionally, we apply the standard data augmentation comprising of random cropping and flipping as described in Goyal et al. (2017).

**iNaturalist**: We use a ResNet-50 trained for 90 epochs with a batch size of 1024. (Note however that for the ResNet-152 experiments in Appendix D.1, we use a batch size of 512.) The base learning rate is 0.4, with cosine learning rate decay. The data augmentation procedure is the same as used in ImageNet.

---

[3]Qualitatively similar results may be achieved with other schedules, which involve training for fewer steps. For example, one may train for 256 epochs with a base learning rate of 0.4, a linear warmup for the first 15 epochs, and a decay of 0.1 at the 96th, 192nd, and 224th epoch.

# D  ADDITIONAL EXPERIMENTS

We present here additional experiments:

(i)  we present a detailed table of results with a more complex base architecture and number of training epochs for ImageNet-LT and iNaturalist;

(ii)  we present results for CIFAR-10 and CIFAR-100 on the STEP profile (Cao et al., 2019) with $\rho = 100$.

(iii)  we present results on synthetic data with varying imbalance ratios.

## D.1  RESULTS WITH MORE COMPLEX BASE ARCHITECTURE

Table 3 presents results when using a ResNet-152, trained for either 90 or 200 epochs, on the larger ImageNet-LT and iNaturalist datasets. Consistent with the findings in Kang et al. (2020), training with a more complex architecture for longer generally yields significant gains. Logit adjustment, potentially when combined with the adaptive margin, is generally superior to baselines with the sole exception of results for ResNet-152 with 200 epochs on iNaturalist.

Table 3: Test set balanced error (averaged over 5 trials) on real-world datasets with more complex base architectures. Employing a ResNet-152 systematically improves all methods' performance, with logit adjustment remaining superior to existing approaches. The final row reports the results of combining logit adjustment with the adaptive margin loss of Cao et al. (2019), which yields further gains on iNaturalist.

| Method | ImageNet-LT | | iNaturalist | | |
| --- | --- | --- | --- | --- | --- |
| | ResNet-50 | ResNet-152 | ResNet-50 90 epochs | ResNet-152 90 epochs | ResNet-152 200 epochs |
| ERM | 53.11 | 53.30 | 38.66 | 35.88 | 34.38 |
| Weight normalisation ($\tau = 1$) (Kang et al., 2020) | 52.00 | 51.49 | 48.05 | 45.17 | 45.33 |
| Weight normalisation ($\tau = \tau^*$) (Kang et al., 2020) | 49.37 | 48.97 | 34.10 | 31.85 | 30.34 |
| Learnable weight scaling (LWS) (Kang et al., 2020) | 52.30 | 49.50 | 34.10 | 30.90 | 27.90 |
| Classifier retraining (cRT) (Kang et al., 2020) | 52.70 | 49.90 | 34.80 | 31.50 | 28.80 |
| Adaptive (Cao et al., 2019) | 52.15 | 53.34 | 35.42 | 31.18 | 29.46 |
| Equalised (Tan et al., 2020) | 54.02 | 51.38 | 38.37 | 35.86 | 34.53 |
| Logit adjustment post-hoc ($\tau = 1$) | 49.66 | 49.25 | 33.98 | 31.46 | 30.15 |
| Logit adjustment post-hoc ($\tau = \tau^*$) | 49.56 | 49.15 | 33.80 | 31.08 | 29.74 |
| Logit adjustment loss ($\tau = 1$) | 48.89 | 47.86 | 33.64 | 31.15 | 30.12 |
| Logit adjustment plus adaptive loss ($\tau = 1$) | 51.25 | 50.46 | 31.56 | 29.22 | 28.02 |

## D.2  PER-CLASS ERROR RATES

Figure 8 breaks down the per-class accuracies on CIFAR-10, CIFAR-100, and iNaturalist. On the latter two datasets, for ease of visualisation, we aggregate the classes into ten groups based on their frequency-sorted order (so that, e.g., group 0 comprises the top $\frac{L}{10}$ most frequent classes). As expected, dominant classes generally see a lower error rate with all methods. However, the logit adjusted loss is seen to systematically improve performance over ERM, particularly on rare classes.

## D.3  RESULTS ON CIFAR-LT WITH STEP-100 PROFILE

Table 4 summarises results on the STEP-100 profile. Here, with $\tau = 1$, weight normalisation slightly outperforms logit adjustment. However, with $\tau > 1$, logit adjustment is again found to be superior (54.80); see Figure 9.

## D.4  RESULTS ON SYNTHETIC DATA WITH VARYING IMBALANCE RATIO

Figure 10 shows results on the synthetic data of §6.1 for varying choice of $\mathbb{P}(y = +1)$. As expected, we see that as $\mathbb{P}(y = +1)$ increases, all methods become equitable in terms of performance. We

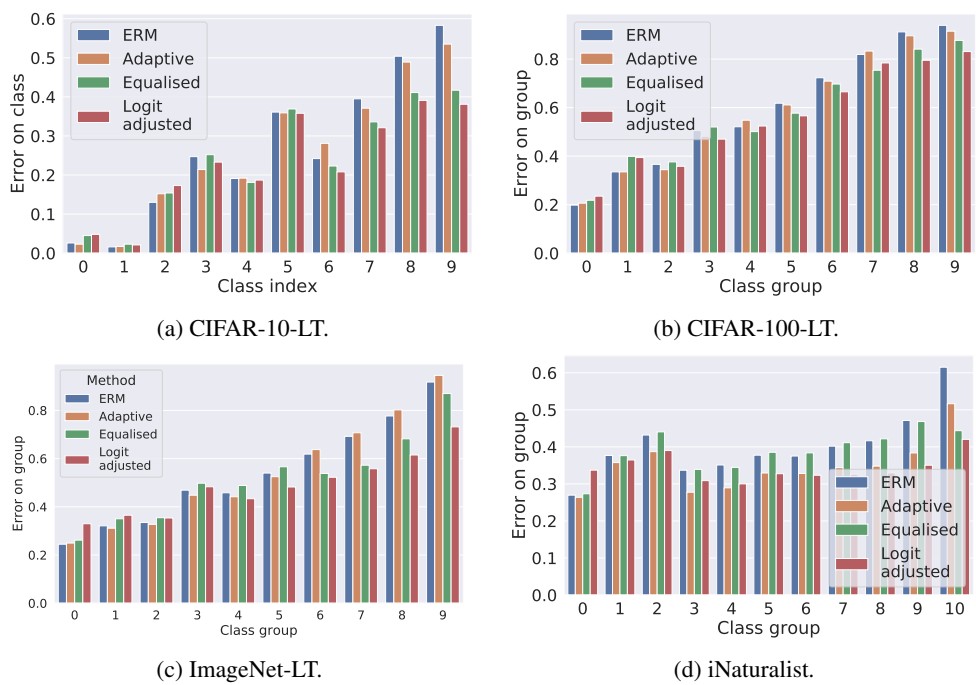

(a) CIFAR-10-LT.

(b) CIFAR-100-LT.

(c) ImageNet-LT.

(d) iNaturalist.

Figure 8: Per-class error rates of loss modification techniques. For (b) and (c), we aggregate the classes into 10 groups. ERM displays a strong bias towards dominant classes (lower indices). Our proposed logit adjusted softmax loss achieves significant gains on rare classes (higher indices).

Table 4: Test set balanced error (averaged over 5 trials) on CIFAR-10-LT and CIFAR-100-LT under the STEP-100 profile; lower is better. On CIFAR-100-LT, weight normalisation edges out logit adjustment. See Figure 9 for a demonstrated that tuned versions of the same outperfom weight normalisation.

| Method | CIFAR-10-LT | CIFAR-100-LT |
|---|---|---|
| ERM | 36.54 | 60.23 |
| Weight normalisation ($\tau = 1$) | 30.86 | 55.19 |
| Adaptive | 34.61 | 58.86 |
| Equalised | 31.42 | 57.82 |
| Logit adjustment post-hoc ($\tau = 1$) | 28.66 | 55.82 |
| Logit adjustment (loss) | 27.57 | 55.52 |

generally find a consistent trend in the relative performance of the various methods, which matches the results in the body.

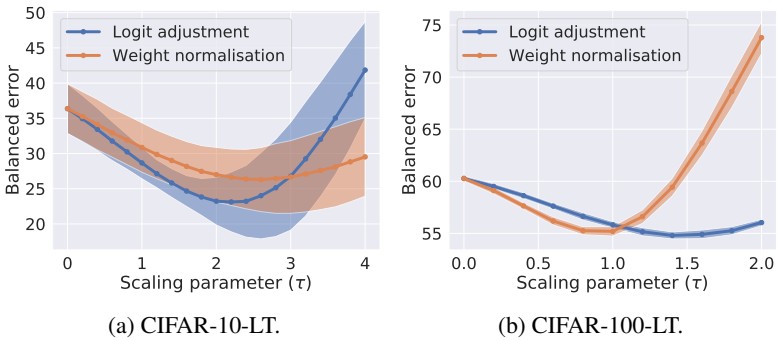

(a) CIFAR-10-LT.          (b) CIFAR-100-LT.

Figure 9: Post-hoc adjustment on STEP-100 profile, CIFAR-10-LT and CIFAR-100-LT. Logit adjustment outperforms weight normalisation with suitable tuning.

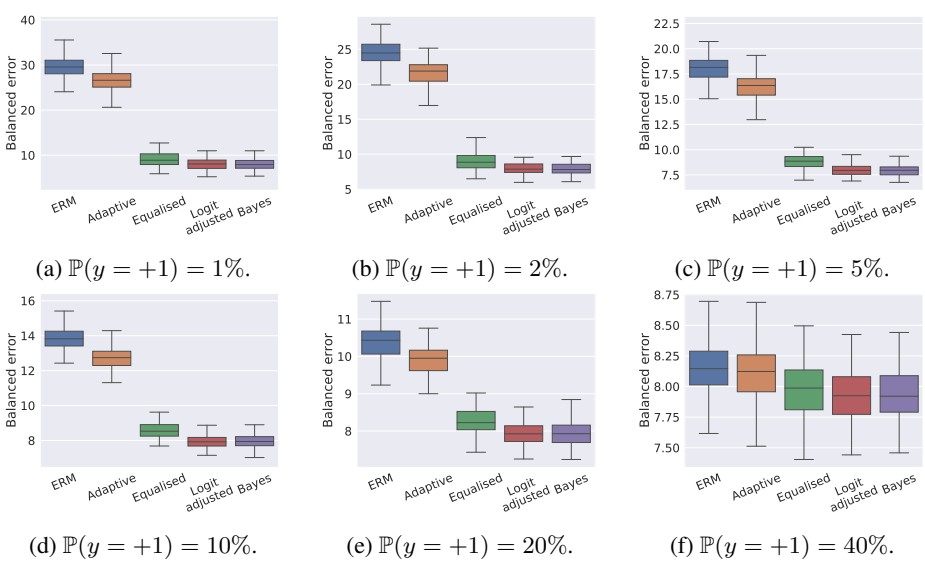

(a) $\mathbb{P}(y = +1) = 1\%$.  (b) $\mathbb{P}(y = +1) = 2\%$.  (c) $\mathbb{P}(y = +1) = 5\%$.

(d) $\mathbb{P}(y = +1) = 10\%$.  (e) $\mathbb{P}(y = +1) = 20\%$.  (f) $\mathbb{P}(y = +1) = 40\%$.

Figure 10: Results on synthetic data with varying imbalance ratio.

# E    DOES WEIGHT NORMALISATION INCREASE MARGINS?

Suppose that one uses SGD with a momentum, and finds solutions where $\|w_y\|_2$ tracks the class priors. One intuition behind normalisation of weights is that, drawing inspiration from the binary case, this ought to increase the classification margins for tail classes. Unfortunately, as discussed below, this intuition is *not* necessarily borne out.

Consider a scorer $f_y(x) = w_y^\top \Phi(x)$, where $w_y \in \mathbb{R}^d$ and $\Phi \colon \mathcal{X} \to \mathbb{R}^d$. The *functional* margin for an example $(x, y)$ is (Koltchinskii et al., 2001)

$$\gamma_f(x, y) \doteq w_y^\top \Phi(x) - \max_{y' \neq y} w_{y'}^\top \Phi(x). \tag{16}$$

This generalises the classical binary margin, wherein by convention $\mathcal{Y} = \{\pm 1\}$, $w_{-1} = -w_1$, and

$$\gamma_{\mathrm{f}}(x, y) \doteq y \cdot w_1^\top \Phi(x) = \frac{1}{2} \cdot \left(w_y^\top \Phi(x) - w_{-y}^\top \Phi(x)\right), \tag{17}$$

which agrees with (16) upto scaling. One may also define the *geometric* margin in the binary case to be the distance of $(x, y)$ from its classifier:

$$\gamma_{g,\mathrm{b}}(x) \doteq \frac{|w_1 \cdot \Phi(x)|}{\|w_1\|_2}. \tag{18}$$

Clearly, $\gamma_{g,\mathrm{b}}(x) = \frac{|\gamma_{\mathrm{f}}(x,y)|}{\|w_1\|_2}$, and so for fixed functional margin, one may increase the geometric margin by minimising $\|w_1\|_2$. However, the same is *not* necessarily true in the multiclass setting, since here the functional and geometric margins do not generally align (Tatsumi et al., 2011; Tatsumi & Tanino, 2014). In particular, controlling each $\|w_y\|_2$ does *not* necessarily control the geometric margin.

## F  BAYES-OPTIMAL CLASSIFIER UNDER GAUSSIAN CLASS-CONDITIONALS

*Derivation of* (12). Suppose

$$\mathbb{P}(x \mid y) = \frac{1}{\sqrt{2\pi}\sigma} \cdot \exp\left(-\frac{\|x - \mu_y\|^2}{2\sigma^2}\right)$$

for suitable $\mu_y$ and $\sigma$. Then,

$$
\begin{aligned}
\mathbb{P}(x \mid y = +1) > \mathbb{P}(x \mid y = -1) &\iff \exp\left(-\frac{\|x - \mu_{+1}\|^2}{2\sigma^2}\right) > \exp\left(-\frac{\|x - \mu_{-1}\|^2}{2\sigma^2}\right) \\
&\iff \frac{\|x - \mu_{+1}\|^2}{2\sigma^2} < \frac{\|x - \mu_{-1}\|^2}{2\sigma^2} \\
&\iff \|x - \mu_{+1}\|^2 < \|x - \mu_{-1}\|^2 \\
&\iff 2 \cdot (\mu_{+1} - \mu_{-1})^\top x > \|\mu_{+1}\|^2 - \|\mu_{-1}\|^2.
\end{aligned}
$$

Now use the fact that in our setting, $\|\mu_{+1}\|^2 = \|\mu_{-1}\|^2$. $\qquad\square$

We now explicate that the class-probability function for the synthetic dataset in §6.1 is exactly in the family assumed by the logistic regression. This implies that logistic regression is well-specified for this problem, and thus can perfectly model $\mathbb{P}(y = +1 \mid x)$ in the infinite sample limit. Note that

$$
\begin{aligned}
\mathbb{P}(y = +1 \mid x) &= \frac{\mathbb{P}(x \mid y = +1) \cdot \mathbb{P}(y = +1)}{\mathbb{P}(x)} \\
&= \frac{\mathbb{P}(x \mid y = +1) \cdot \mathbb{P}(y = +1)}{\sum_{y'} \mathbb{P}(x \mid y') \cdot \mathbb{P}(y')} \\
&= \frac{1}{1 + \frac{\mathbb{P}(x|y=-1)\cdot\mathbb{P}(y=-1)}{\mathbb{P}(x|y=+1)\cdot\mathbb{P}(y=+1)}}.
\end{aligned}
$$

Now,

$$
\begin{aligned}
\frac{\mathbb{P}(x \mid y = -1)}{\mathbb{P}(x \mid y = +1)} &= \exp\left(\frac{\|x - \mu_{+1}\|^2 - \|x - \mu_{-1}\|^2}{2\sigma^2}\right) \\
&= \exp\left(\frac{\|\mu_{+1}\|^2 - \|\mu_{-1}\|^2 - 2 \cdot (\mu_{+1} - \mu_{-1})^\top x}{2\sigma^2}\right) \\
&= \exp\left(\frac{-(\mu_{+1} - \mu_{-1})^\top x}{\sigma^2}\right).
\end{aligned}
$$

Thus,

$$\mathbb{P}(y = +1 \mid x) = \frac{1}{1 + \exp(-w_*^\top x + b_*)},$$

where $w_* = \frac{1}{\sigma^2} \cdot (\mu_{+1} - \mu_{-1})$, and $b_* = \log \frac{\mathbb{P}(y=-1)}{\mathbb{P}(y=+1)}$. This implies that a sigmoid model for $\mathbb{P}(y = +1 \mid x)$, as employed by logistic regression, is well-specified for the problem. Further, the bias term $b_*$ is seen to take the form of the log-odds of the class-priors per (8), as expected.

