# OpenReview forum: "Long-tail learning via logit adjustment"
_ICLR.cc/2021/Conference — ICLR 2021 Spotlight_

### Official Review · AnonReviewer4 · 2020-10-21
**This papers presents a general loss function for long-tail classification with several previous work as its special cases.**

**Rating:** 6
**Confidence:** 4

**Review:**

This papers presents a general loss function for long-tail classification with several previous work as its special cases.

This is a well-written paper, and the results are impressive. The approach builds upon prior work and a general framework is presented. The proposed approaches are eveluated on several commonly used datasets and show some improvements.

My one major technical concern are as follows:
1. The originality of this paper is not very high since the proposed framework and its components are not novel (there might be some minor novelty such as the fisher consistency property of the objective);
2. Regarding the post-hoc logit adjustment, I am supposing it is sensitive to $\pi_y^\tau$, which is not very much similar with weight normalization;
3. For the balanced error, I am interested in why it is supposed to a better performance measure, given that the test data distribution is uniform (as per datasets used in experiments);
4. In the experiments, e.g., Table 2, in my humble opinion, some of resfults for comparison methods are incorrect. Since prior work (including Weight normalisation, Adaptive, Equalised) report Top-1 classification error, instead of the balanced error. Hence, I guess that the comparison is not fair at all.

========== after reading the authors feedback =========

Thanks the authors for addressing my concerns and I am convinced that this work is very much different from prior literature. In addition, the evaluation metrics are correct in the studied problem setup. Based on that, I would like to raise my score from 5 to 6.

---

> ### Author Response · Authors · 2020-11-17
> **Response to R4**
>
> Thanks for your encouraging review, and the detailed feedback!
>
> *# The originality of this paper is not very high since the proposed framework and its components are not novel (there might be some minor novelty such as the fisher consistency property of the objective);*
>
> Compared to prior work, we:
> 1) Establish limitations with some existing techniques both theoretically (Sec 4.2, Sec 5.2) and empirically (Sec 2.1, 6.1)
> 2) Provide a unified pairwise margin loss (Eqn 10) that captures existing techniques as special cases, and highlights the favourable statistical grounding of our proposed loss (Theorem 2)
> 3) Empirically demonstrate on modern long-tail benchmarks that logit adjustment techniques can be superior or competitive with existing techniques.
>
> *# Regarding the post-hoc logit adjustment, I am supposing it is sensitive to πyτ, which is not very much similar with weight normalization*
>
> Please note that Table 2 only reports results for logit adjustment with τ = 1, which requires no tuning; these results are already generally superior or competitive to existing techniques.
>
> Further, Figure 3 shows how performance varies with τ for both weight normalisation and logit adjustment. The two methods have similar trends, and in particular logit adjustment does not have greater sensitivity to the choice of τ.
>
> *# For the balanced error, I am interested in why it is supposed to a better performance measure, given that the test data distribution is uniform (as per datasets used in experiments)*
>
> The reviewer is correct that the benchmarks we report results on have a balanced *test* label distribution. However, this need not be true in real-world applications. e.g., if building a predictor for whether a patient has a disease affecting 1% of the population, it is unlikely that the test set will comprise a balance of patients with and without the disease.
>
> In settings with imbalanced test data, the balanced error is more informative than the misclassification error. Consider a trivial classifier that predicts every patient as not having the disease: this would get a misclassification error of 1% (which appears artificially good), but a balanced error of 50% (which correctly deems the classifier to be no better than random guessing).
>
> Further to the above, note that since the *training* label distribution is imbalanced, optimising for the balanced versus misclassification error here can strongly impact performance on rare labels. For more discussion on the merits of balanced error under class imbalance, please see [Brodersen et al., 2010], [Menon et al., 2013], [Koyejo et al., 2014].
>
> *# In the experiments, e.g., Table 2, in my humble opinion, some of resfults for comparison methods are incorrect. Since prior work (including Weight normalisation, Adaptive, Equalised) report Top-1 classification error, instead of the balanced error. Hence, I guess that the comparison is not fair at all.*
>
> Please note that the balanced accuracy and top-1 accuracy are *equivalent* for these datasets, and so the comparisons are fair. As the reviewer notes, the test sets for these benchmarks are all balanced. Consequently, P(y) = 1/L for each y. Thus, the balanced accuracy in (2) is equivalent to weighting each per-class accuracy by P(y), which is exactly the standard accuracy. We have clarified this prior to the discussion of results.
>
> To further clarify, employing the balanced error also compares the top-1 prediction of the model (i.e., highest scoring logit) against the true label. We have clarified this following Equation 2.

---

### Official Review · AnonReviewer1 · 2020-10-26
**Simple algorithm for long-tail learning with Fisher consistency**

**Rating:** 7
**Confidence:** 3

**Review:**


Summary:
This paper proposes an unifying statistical framework for imbalanced or long-tailed data, where the number of samples for some of the classes may be extremely small compared with other classes. Previous methods work empirically well, but was not consistent, meaning that even in the infinite sample limit, the minimiser doesn't result in a minimal balanced error. The paper proposes a framework based on logit adjustment in two ways: a post-hoc logit adjustment way and another way that injects the logit adjustment to the softmax cross-entropy loss function directly. The paper shows that they are both consistent. Experiments show that the proposed methods work better than previous methods.

Pros:
I feel the paper is very well organized and the story of the paper is clear.  The paper explains the issues of recent methods such as weight normalization and loss modification methods in long-tail learning, and propose a simple statistical formulation with consistency guarantee. It gives further theoretical insights for the binary classification case.

The synthetic data experiments are well designed, and it gives the reader a better understanding of the behavior of the proposed method. The proposed method matches the Bayes optimal decision boundary, while previous methods and naive ERM seem to be worse, demonstrating the characteristics of proposed and previous methods. The ablation study of the scaling parameter shows how a default value of 1 is already good enough in most cases, but can be tuned for further performance gains.

Cons:
The ImageNet-LT results were shown in the Appendix, but the proposed Post-hoc correction seems to be slightly worse than weight normalization. I feel readers will be more comfortable to have this in the main paper.

Other comments:
What does the gray-colored highlights in Table 2 mean? If it means the best performing method, then for CIFAR-10-LT, weight normalisation should be highlighted instead of logit adjustment loss.

It would be nicer to explain that "the proofs for the two theorems are in the supplementary" in page 6. Since the supplementary was in a separate PDF file, I didn't notice this when I first read this paper.

On page 8, it says "See Appendix D.4 for a plot on ImageNet-LT", but this should be D.2 instead.

*******************
After rebuttal period: thank you for answering my questions and for updating the paper. I still think it is a good paper and would like to keep my score.

---

> ### Author Response · Authors · 2020-11-17
> **Response to R1**
>
> Thanks for your encouraging review, and the detailed feedback!
>
> *# The ImageNet-LT results were shown in the Appendix, but the proposed Post-hoc correction seems to be slightly worse than weight normalization. I feel readers will be more comfortable to have this in the main paper.*
>
> Note that the gains from weight normalisaton in this case were shown in Table 2. We agree however that including the plots in the body makes this clearer, and have updated Figure 3 and 4 accordingly.
>
> *# What does the gray-colored highlights in Table 2 mean? If it means the best performing method, then for CIFAR-10-LT, weight normalisation should be highlighted instead of logit adjustment loss.*
>
> The highlighted cells indicate the best performing method. We have corrected the highlighting on CIFAR-10-LT.
>
> Please note that Table 2 originally compared weight normalisation with the *optimal* tau against logit adjustment with τ = 1 (which is somewhat unfair to our technique). For CIFAR-10-LT, with cross-validation to select τ, post-hoc logit adjustment achieves a superior performance of 18.73%. We have added a discussion of this in the “Results and analysis” section. We have also added results for weight normalisation with τ = 1 into Table 2, for clarity.
>
> *# It would be nicer to explain that "the proofs for the two theorems are in the supplementary" in page 6.*
>
> We have added a comment to this effect.

---

### Official Review · AnonReviewer3 · 2020-10-27
**A novel statistical framework for long-tail learning that pursues Fisher consistent for minimizing the balanced error.**

**Rating:** 8
**Confidence:** 4

**Review:**

[Summary]

This paper provides a statistical framework for long-tail learning by revisiting the idea of logit adjustment based on the label frequencies. The proposed framework then yields two variant techniques that follow the paradigm of weight normalization or loss modification. Compared with the existing methods, the proposed methods are generalized and Fisher consistent for minimizing the balanced error. Finally, empirical results on four real-world datasets validate the effectiveness and statistical grounding of the proposed methods.

[Pros]
-	The idea of this paper is novel and interesting. The proposed framework is proved to be Fisher consistent for minimizing the balanced error and generalizes several recent methods for long-tail learning. Meanwhile, some insights about the logit adjustment technique are also revealed to help understanding.
-	The experiments are sufficient and supportive to validate the effectiveness and statistical grounding of the proposed methods.
-	The paper is well organized, which makes it easy to grasp the core idea.

[Cons]
-	From the left panel of Fig.2 it seems that the error bar of logit adjusted is even thinner than that of the Bayes predictor. It would be better to provide some explanation.
-	It seems that the proposed framework could work well with linear classifiers. Does it also applies to other classifiers such as cosine classifier?
-	Captions of tables should be put above the table contents.
-	Besides, there are some grammatical errors and typos to be corrected. Some are lists as follows:
  1)	Page 3, ‘an non-decreasing transform’ should be ‘a non-decreasing transform’;
  2)	Page 4, ‘which overcome’ should be ‘which overcomes’;
  3)	Page 5, ‘has negative score’ should be ‘has a negative score’.
  4)	Page 5, ‘another label with positive score’ should be ‘another label with a positive score’.

---

> ### Author Response · Authors · 2020-11-17
> **Response to R3**
>
> Thanks for your encouraging review, and the detailed feedback!
>
> *# From the left panel of Fig.2 it seems that the error bar of logit adjusted is even thinner than that of the Bayes predictor. It would be better to provide some explanation.*
>
> Thanks for spotting this: the original Figure 2 was erroneously reported for a smaller (10) number of trials, and the Bayes result was affected by a single trial having a larger error. We have updated it to be the result for 100 trials, and the error bars are now commensurate.
>
> *# It seems that the proposed framework could work well with linear classifiers. Does it also applies to other classifiers such as cosine classifier?*
>
> The proposed framework should be applicable whenever the underlying model produces reasonable probability estimates. Classifiers based on normalised embeddings can typically achieve this, and so we do not foresee difficulties here. Exploring this would be of interest in future work.
>
> *# Captions of tables should be put above the table contents.*
>
> We have updated this, and fixed all typos.

---

### Official Review · AnonReviewer2 · 2020-10-28
**Review of AnonReviewer2**

**Rating:** 8
**Confidence:** 4

**Review:**

Summary:
This well-written paper re-visits the idea of logit adjustment to tackle long-tailed problems. The paper begins by setting up a statistical framework and use that to deliver two ways of realizing the logits adjustment effectively. They further prove the potential of such an approach by benchmarking it with several related baselines on both synthetic and natural long-tailed datasets.

###########################################################
+ves:
+ The paper motivates the proposed method very well, by exposing the cases of failures of certain existing approaches and addressing those shortcomings in the proposed method.
+ The explanation about how the proposed methods standout - that is post-hoc logit adjustment with respect to the weight normalization and logit adjusted loss with respect to the LDAM loss is clear and well written.
+ The proposed methods are very versatile as it can both be incorporated into the training, used after the training, and in combination with each other.
+ The experiments and the analysis are both comprehensive, and the paper has a nice technical depth to it.
+ The paper is very well-written.

###########################################################
Concerns/Potential Improvements:
- There is no justification as to why results on synthetic datasets are provided only for the imbalance ratio of 100 while it is a community norm to benchmark on a range of imbalance ratios (typically 100, 50, 20, 10, 1)
- Weight normalization has a term multiplied to the logits and the proposed post-hoc logit adjustment has a term added to the argmax of the logits. Therefore both the terms are independent of each other. It would be interesting to see how both of these methods work in conjunction. Does it improve the results or the cons of one just penalizes the pros of another?

###########################################################
Minor editorial/typo issues:
- For post-hoc logit adjustment, the paper provides a separate subsection named “COMPARISON TO POST-HOC WEIGHT NORMALISATON”. It would have been nice to see something similar for logit adjusted loss. The content is already there in the paper, and just has to be modified a bit to stand out separately.
- Near equation 9, the paper mentions that “(8) immediately suggests two means of optimizing for the balanced error” and goes on to provide the two methods. This was not very evident from (8). Some description to link would have been nice.

POST-REBUTTAL:

I thank the authors for their response. I am happy with the responses to my concerns/questions, and retain my decision of Accept.

---

> ### Author Response · Authors · 2020-11-17
> **Response to R2**
>
> Thanks for your encouraging review, and the detailed feedback!
>
> *# There is no justification as to why results on synthetic datasets are provided only for the imbalance ratio of 100 while it is a community norm to benchmark on a range of imbalance ratios (typically 100, 50, 20, 10, 1)*
>
> We have now included in Appendix D.3 experiments with multiple imbalance ratios. The trends are consistent with the case of imbalance ratio 100, as reported in the original submission. As expected, as the imbalance ratio increases, so does the gap between baselines and the Bayes/logit adjusted predictor.
>
>
> *# Weight normalization has a term multiplied to the logits and the proposed post-hoc logit adjustment has a term added to the argmax of the logits. Therefore both the terms are independent of each other. It would be interesting to see how both of these methods work in conjunction. Does it improve the results or the cons of one just penalizes the pros of another?*
>
> We have experimented with combining post-hoc weight normalisation and logit adjustment. However, we generally have not found significant gains from doing so. This could be because both techniques have the effect of increasing the logits of rare classes, albeit in different ways. Further exploring means of combining these techniques could definitely be an interesting direction for future study.
>
>
> *# For post-hoc logit adjustment, the paper provides a separate subsection named “COMPARISON TO POST-HOC WEIGHT NORMALISATON”. It would have been nice to see something similar for logit adjusted loss. The content is already there in the paper, and just has to be modified a bit to stand out separately.*
>
> We have followed the reviewer’s suggestion, and added a new subsection heading 5.2 for contrasting against loss modification techniques.
>
> *# Near equation 9, the paper mentions that “(8) immediately suggests two means of optimizing for the balanced error” and goes on to provide the two methods. This was not very evident from (8). Some description to link would have been nice.*
>
> We have updated the text following (8) to further clarify how these techniques are arrived at.

---

### Author Response · Authors · 2021-01-25
**Regarding Figure 4**

There was an earlier comment regarding the iNaturalist results in Figure 4, which breaks down the errors by classes Many/Medium/Few samples. We can confirm that our ERM implementation achieves an error of 27.9% on the Many bucket, and that it has the expected trend of being worse on the Medium and Few samples. There seems to have been a mixup in the plot, which shall be corrected in the final version.

---

### Decision · Program_Chairs · 2021-01-07
**Final Decision**

**Decision:**

Accept (Spotlight)

**Comment:**

This paper got 3 acceptance and 1 marginally below the threshold. After the rebuttal, the rating was raised to above the threshold. All the reviewers are positive about this submission. They agree that the method proposed in the submission is novel, the experiments are comprehensive and convincing. AC agrees and recommend acceptance.